# Stretchable chiral pockets for palladium-catalyzed highly chemo- and enantioselective allenylation

Yuchen Zhang[1], Xue Zhang[2 ✉] & Shengming Ma [1,2 ✉]

Pyrazolones are a vital class of heterocycles possessing various biological properties and much attention is paid to the diversified synthesis of enantiopure pyrazolone derivatives. We describe here the development of diphenylphosphinoalkanoic acid based chiral bisphosphine ligands, which are successfully applied to the palladium-catalyzed asymmetric allenylation of racemic pyrazol-5-ones. The reaction affords C-allenylation products, optically active pyrazol-5-ones bearing an allene unit, in high chemo- and enantioselectivity, with **DACH-ZYC-Phos-C1** as the best ligand. The synthetic potential of the C-allenylation products is demonstrated. Furthermore, the enantioselectivity observed with **DACH-ZYC-Phos-C1** is rationalized by density functional theory studies.

---

[1] Laboratory of Molecular Recognition and Synthesis, Department of Chemistry, Zhejiang University, Hangzhou, Zhejiang, P. R. China. [2] State Key Laboratory of Organometallic Chemistry, Shanghai Institute of Organic Chemistry, Chinese Academy of Sciences, Shanghai, P. R. China. ✉email: xzhang@sioc.ac.cn; masm@sioc.ac.cn

Pyrazolones are a vital class of heterocycles possessing various biological properties[1]. Of particular interest are the pyrazolone derivatives with a chiral center (Fig. 1a): compound **A** and its derivatives were proven to be orally active growth hormone secretagogues[2,3]; compound **B** was used as a modulator of TRPM8;[4] compound **C** shows remarkable anti-tumor activity related to lung, colon, and breast cancer cell[5,6]. Thus, much attention has been paid to the diversified synthesis of bioactive enantiopure pyrazolone derivatives[7–27]. Along this line, enantioselective allylation of pyrazolones has been established[9–12]. For allene derivatives, enantioselective spirocyclizations of pyrazolones with 2,3-allenyl acetates have been developed by applying chiral phosphine or transmetal/chiral ligand catalysis (Fig. 1b)[13,14]. So far, there is no report on enantioselective 2,3-allenylation of pyrazolones, which is very attractive owing to the unique biological[28–30] and chemical properties[31–37] of the allene unit and pyrazolone moiety.

In this work, inspired by conformationally flexible alkyl chain in the Feng's ligands, we develop the enantioselective allenylation of pyrazolones by fine tuning of the Trost ligands, which leads to the development of a class of stretchable chiral pocket (Fig. 1c).

## Results and discussion

Initially, we studied the enantioselective allenylation of pyrazol-5-one **2a** with benzyl 2-butylbuta-2,3-dienyl carbonate **1a** with some typical and commercially available chiral ligands at 30 °C in CHCl₃ (Fig. 2). With **L1** and **L8** as the ligand, there was a serious competing reaction forming N-allenylation product **4aa**. Further screening led to the observation that **L2–L7** were sluggish for this allenylation. In the reaction with Trost ligand **L9**, no product was observed, while Trost ligand **L10** afforded the C-allenylation product (S)-**3aa** in 73% yield and 50% ee; interestingly, Trost ligand **L11** could give (S)-**3aa** in 84% yield and 75% ee. With **L11** as the ligand, subsequent investigation on the solvent effect showed that toluene was the best one, affording 85% yield (S)-**3aa**

with 84% ee (Fig. 2, entries 1–5). After optimizing the temperature and concentration (Fig. 2, entries 6–11), the reaction at 0.02 M could give (S)-**3aa** in 92% yield and 85% ee, together with a trace amount of **4aa** at 60 °C (Fig. 2, entry 11).

Obviously, a higher enantioselectivity was desired and the success of conformationally flexible alkyl chain in the Feng's ligands[38–41] has caught our attention (Fig. 3a). It was reasoned that the rigid aryl linker in the Trost ligands may be replaced with a flexible alkyl linker for a class of stretchable chiral pockets seeking higher enantioselectivity. Thus, the **ZYC-Phos** ligands were designed and readily synthesized by amidation of diphenylphosphino alkanoic acids, which were prepared by nucleophilic substitution of ethyl chloroalkanoate or chloroacetic acid (Fig. 3b) for further optimization[42–45].

With the ligands in hand, we conducted the allenylation with **DACH-ZYC-Phos-C2**. The reaction afforded (S)-**3aa** in 85% ee, albeit with a low conversion (14% yield with 67% recovery of **1a**) (Table 1, entry 1). Further optiminization was performed: the reactions at 60 °C and 80 °C afforded (S)-**3aa** in 75 and 80% yields, respectively, with a similar ee together with 4% of the N-allenylation product **4aa** (Table 1, entries 2 and 3). Next, we investigated the effect of concentration (Table 1, entries 4–6) and observed that the ee of (S)-**3aa** was improved to 88% when the concentration of the reaction was 0.02 M. The ee value of (S)-**3aa** was similar with 5.0 mol% of the ligand (Table 1, entry 7). **DADPE-ZYC-Phos-C2**, **DACH-ZYC-Phos-C3**, and **DACH-ZYC-Phos-C1** (Table 1, entries 8–10) were then screened and **DACH-ZYC-Phos-C1** was found to afford (S)-**3aa** in 79% yield and 90% ee as the chiral ligand with 2% of N-allenylation product **4aa**. Solvent screening (Table 1, entries 11–13) led to the conclusion that toluene was the best choice, exclusively affording (S)-**3aa** in 100% yield and 95% ee. Using toluene as solvent, **DACH-ZYC-Phos-C2** and **C3** were re-tested (Table 1, entries 14 and 15). Thus, the optimal conditions were redefined as follows: allene **1**, pyrazolone **2** (1.2 equiv),

**Fig. 1 Background and stretchable ligands for asymmetric allenylation of pyrazolones. a** Selected examples of biologically active pyrazolones with a chiral center. **b** Asymmetric spirocyclization of pyrazolones with 2,3-allenyl acetates. **c** Stretchable chiral pocket for asymmetric allenylation. EWG electron-withdrawing group, TMS trimethylsilyl, LG leaving group.

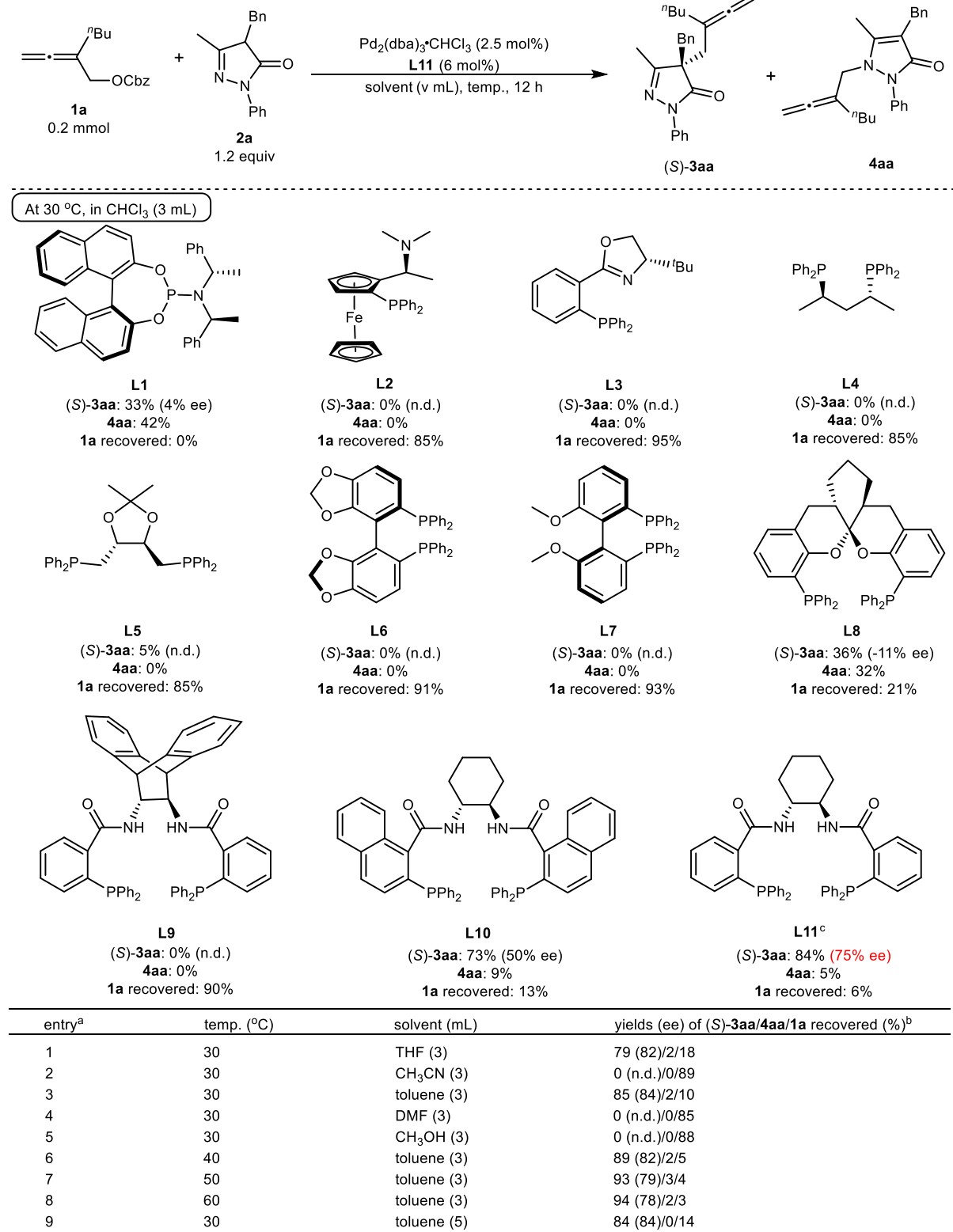

| entry[a] | temp. (°C) | solvent (mL) | yields (ee) of (S)-3aa/4aa/1a recovered (%)[b] |
|---|---|---|---|
| 1 | 30 | THF (3) | 79 (82)/2/18 |
| 2 | 30 | CH₃CN (3) | 0 (n.d.)/0/89 |
| 3 | 30 | toluene (3) | 85 (84)/2/10 |
| 4 | 30 | DMF (3) | 0 (n.d.)/0/85 |
| 5 | 30 | CH₃OH (3) | 0 (n.d.)/0/88 |
| 6 | 40 | toluene (3) | 89 (82)/2/5 |
| 7 | 50 | toluene (3) | 93 (79)/3/4 |
| 8 | 60 | toluene (3) | 94 (78)/2/3 |
| 9 | 30 | toluene (5) | 84 (84)/0/14 |
| 10 | 30 | toluene (10) | 75 (86)/0/25 |
| 11[d] | 60 | toluene (10) | 92 (85)/0/0 |

**Fig. 2 The selected data of optimization with some commercially available chiral ligands.** [a]Reaction conditions: **1a** (0.2 mmol), **2a** (1.2 equiv), Pd₂(dba)₃•CHCl₃ (2.5 mol%), and **L1**-**4** (12 mol%) or **L5**-**11** (6 mol%) unless otherwise noted. [b]The yields of (S)-**3aa** and **4aa** as well as the recovery of **1a** were determined by the ¹H NMR analysis of the crude product using mesitylene as the internal standard and the ee of isolated (S)-**3aa** was determined by chiral HPLC. [c]The reaction time was 4 h. [d]**L11** (5 mol%) was used and the reaction time was 3 h. Cbz benzyloxycarbonyl, THF tetrahydrofuran, DMF N,N-dimethylformamide, n.d. not determined.

**Fig. 3 Concept and synthesis of the ZYC-Phos ligands. a** The concept of **ZYC-Phos** ligands from the Feng's ligands and the Trost ligands. **b** The synthesis of **ZYC-Phos** ligands. EDCI 1-ethyl-(3-dimethylaminipropyl)carbodiimide hydrochloride, NHS N-hydroxysuccinimde, DMSO dimethyl sulfoxide, DCM dichloromethane, DACH (1R,2R)-1,2-diaminocyclohexane, DADPE (1R,2R)-1,2-diphenyl-1,2-diaminoethane, DMAP 4-dimethylaminopyridine.

Pd₂(dba)₃•CHCl₃ (2.5 mol%), and **DACH-ZYC-Phos-C1** (5.0 mol %) in toluene (0.02 M) at 60 °C (Table 1, entry 11).

The scope of benzyl buta-2,3-dienyl carbonates **1** was then examined with pyrazol-5-one **2a**. For R¹, n-butyl, methyl, n-octyl, TMS, and phenyl all worked well with an excellent yield and ee (Table 2, entries 1–5). However, when R¹ is H, the ee value of (R)-**3fa** dropped to 78% with 86% yield, indicating a serious steric effect (Table 2, entry 6). The scope of pyrazol-5-ones **2** was then examined with benzyl 2-(n-butyl)buta-2,3-dienyl carbonate **1a**. The reaction of 4-(ortho-, meta- or para-methylbenzyl)pyrazol-5-ones **2b**–**2d** with **1a** afforded (R)-**3ab**~(S)-**3ad** in similar yields and enantioselectivity (Table 2, entries 7–9), suggesting that the location of the substituent on the aryl group in the 4-benzyl unit has very limited effect for enantioselectivity. Some synthetic versatile functionalities, such as Cl, Br, F, CN, CF₃, MeO, and NO₂, survived well in the reaction of **1a** and pyrazol-5-ones **2**, affording (S)-**3ae**~(S)-**3ak** in 86–97% yields and 94–98% ee (Table 2, entries 10–16). The absolute configuration was determined by the X-ray single crystal diffraction analysis of (S)-**3ak**

(Fig. 4a). Furthermore, 4-ethyl, 4-allyl, and 4-(α-naphthylmethyl) pyrazol-5-ones **2l**–**2n** were exercisable with 87–96% yields and 93–97% ee (Table 2, entries 17, 18, 20). As expected, enantiomer (R)-**3am** was obtained in 95% yield and 94% ee by using ent-**DACH-ZYC-Phos-C1** as the ligand (Table 2, entry 19). It should be noted that the difference between the ee values of (R)-**3ao** and (R)-**3ap** suggested that coordination of the hydroxyl oxygen may affect the enantioselectivity (Table 2, entries 21 and 22). Furthermore, for R³ being phenyl, the ee value of (R)-**3at** was only 36% with 62% yield (Table 2, entry 23). 3-Ethyl, 3-isopropyl, and 3-phenyl pyrazol-5-ones **2q**–**2s** also worked smoothly to offer the products in 91–>99% yields and 95–97% ee (Table 2, entries 24–26). The reaction of N-cyclohexyl pyrazol-5-one **2u** and **1b** afforded (S)-**3bu** in 84% yield with 95% ee while the reaction of N-benzyl pyrazol-5-one **2v** and **1b** afforded (S)-**3bv** in 82% yield with 90% ee (Table 2, entries 27 and 28).

The reaction of **1a** and **2m** could be conducted on gram scale facily, affording 1.20 g of (S)-**3am** in a similar yield and enantioselectivity (Fig. 4b–i). Subsequently, an allenic Pauson–Khand

**Table 1 The effect of temperature, concentration, ligands, and solvents on the asymmetric allenylation of pyrazolone 2a with allene 1a[a].**

| Entry | ZYC-Phos | Solvent (v) | Yield (ee) of (S)-3aa/4aa/1a recovered (%)[b] |
|---|---|---|---|
| 1[c] | **DACH-ZYC-Phos-C2** | CHCl₃ (3) | 14 (85)/trace/67 |
| 2 | **DACH-ZYC-Phos-C2** | CHCl₃ (3) | 75 (87)/4/6 |
| 3[d] | **DACH-ZYC-Phos-C2** | CHCl₃ (3) | 80 (86)/4/0 |
| 4[e] | **DACH-ZYC-Phos-C2** | CHCl₃ (1.5) | 67 (82)/7/4 |
| 5[e] | **DACH-ZYC-Phos-C2** | CHCl₃ (5) | 78 (87)/3/4 |
| 6[f] | **DACH-ZYC-Phos-C2** | CHCl₃ (10) | 70 (88)/2/12 |
| 7 | **DACH-ZYC-Phos-C2** | CHCl₃ (10) | 76 (89)/2/17 |
| 8 | **DADPE-ZYC-Phos-C2** | CHCl₃ (10) | 79 (79)/5/8 |
| 9[g] | **DACH-ZYC-Phos-C3** | CHCl₃ (10) | 84 (76)/7/0 |
| 10 | **DACH-ZYC-Phos-C1** | CHCl₃ (10) | 79 (90)/2/10 |
| 11[h] | **DACH-ZYC-Phos-C1** | Toluene (10) | 100 (95)/0/0 |
| 12 | **DACH-ZYC-Phos-C1** | THF (10) | 92 (83)/3/0 |
| 13 | **DACH-ZYC-Phos-C1** | CH₃CN (10) | 23 (n.d.)/4/60 |
| 14 | **DACH-ZYC-Phos-C2** | Toluene (10) | 87 (96)/0/11 |
| 15 | **DACH-ZYC-Phos-C3** | Toluene (10) | 90 (87)/0/0 |

[a]Reaction conditions: **1a** (0.2 mmol), **2a** (1.2 equiv), Pd₂(dba)₃•CHCl₃ (2.5 mol%), and ligand (6 mol% for entries 1–6; 5 mol% for entries 7–13) unless otherwise noted. [b]The yields of (S)-**3aa** and **4aa** as well as the recovery of **1a** were determined by the ¹H NMR analysis of the crude product using mesitylene as the internal standard and the ee of isolated (S)-**3aa** was determined by chiral HPLC. [c]At 30 °C for 20 h. [d]At 80 °C. [e]The reaction time was 14 h. [f]The reaction time was 16 h. [g]The reaction time was 12.5 h. [h]The reaction was conducted on 0.5 mmol scale. Cbz benzyloxycarbonyl, THF tetrahydrofuran.

annulation of (S)-**3am** was established in 55% yield and 99% ee to afford tricyclic product (3a*S*,5 *S*)-**5am** containing a bicyclic motif with anti-tumor activity[46] (see compound **C** in Fig. 1a). The annulation reaction of (S)-**3da** with *o*-(propen-2-yl)phenol could easily construct a seven-membered heterocycle in product (*S*,*Z*)-**6da** (72% yield, 95% ee and > 20/1 *Z/E*)[47]. The allene unit in (S)-**3da** also could be transformed into a 2-propynyl side chain in (S)-**7da** upon treatment with TiCl₄.

A possible mechanism for the allenylation of pyrazolone is shown in Fig. 5a[48–50]. To better understand the superior enantioselectivity and the advantage of **DACH-ZYC-Phos-C1** in the allenylation of pyrazolone, we have conducted X-ray single crystal diffraction studies: firstly, Pd(II)-**DACH-ZYC-Phos-C1** complex (**III**) was prepared by the reaction of PdCl₂ and **DACH-ZYC-Phos-C1** with excess amount of bases in toluene (Fig. 5b). Pd(II)-DACH-Phenyl Trost ligand complex (**IV**) was obtained from the reaction of Pd(OAc)₂ with DACH-Phenyl Trost ligand with excess amount of bases in THF[51]. Both complexes were then recrystalized from CHCl₃/*n*-hexane to afford single crystals suitable for the X-ray diffraction study. It was obvious that the angle (P1-Pd1-P2) and the distance between P1 and P2 in **III** are greater than those in **IV** (113.21° vs 102.19°, 3.773 Å vs 3.489 Å). However, this complex **III** failed to catalyze the enantioselective allenylation under the standard conditions or with AcOH as the protic additive (Fig. 5d), indicating that the complex **III** is not really a catalytically active species but provides a stable coordination mode for the palladium catalysis. The X-ray crystal structures of **III** and **IV** were then taken as the starting geometries for all the following calculations involving these complexes by restoring the ligand's amide N-H.

DFT calculations were performed on the enantioselectivity determining C−C bond formation step of the reaction of benzyl buta-2,3-dienyl carbonates **1b** with pyrazol-5-one **2a** catalyzed by the **DACH-ZYC-Phos-C1**-ligated palladium catalyst (see computational methods in the Supplementary Information for details). The reported energies are Gibbs energies that incorporate the effect of the toluene solvent.

Figure 6 shows the optimized structures and relative free energies of the competing transition states basing on the endo-methylene-π-allyl palladium complexes[52] (see the Supplementary Information for the other less favorable transition structures with exo-methylene-π-allyl palladium complexes). These transition structures are denoted as **TS_left_*Si*_a/b**, **TS_left_*Re*_a/b**, and **TS_right_*Si***, and **TS_right_*Re***, separately (left/right indicates that the side of methylene moiety referring to the π-allyl Pd unit). Among them, **TS_left_*Si*_a** with the *Si*-face attack of **2a** anion, is found to be the most favorable one, which is consistent with the dominant formation of *S*-products observed in the experimental studies.

The ligand's right-side amide N-H is available for hydrogen bonding to either the carbonyl oxygen or the hydrazine nitrogen of the approaching **2a** anion. The structures of **TS_left_*Si*_a** and **TS_left_*Si*_b**, leading to the formation of the allenylation product with *S* configuration, are stabilized by the hydrogen bonding forming between the ligand's N-H with the carbonyl oxygen and the hydrazine nitrogen, separately. In the structure of **TS_left_-*Si*_a**, the H···O distance is 1.81 Å, and the bond angle of N−H···O is 175.0°, which are consistent with the common hydrogen bonding parameters[53]. Based on the electron density at the bond critical point, the hydrogen binding energies (BE) were calculated to analyze the strength of the hydrogen bonding[54]. The bond energy of the hydrogen bonding in **TS_left_*Si*_a** was estimated to be about 8.1 kcal/mol. Although the influence of the π/π interactions between the aromatic rings of the ligand and N-Ph

**Table 2 The scope of allenes and pyrazol-5-ones[a].**

1
0.5 mmol

R[1] = $^n$Bu, **1a**; Me, **1b**
$^n$C$_8$H$_{17}$, **1c**; TMS, **1d**
Ph, **1e**; H, **1f**

2
1.2 equiv

Pd$_2$(dba)$_3$•CHCl$_3$ (2.5 mol%)
**DACH-ZYC-Phos-C1** (5.0 mol%)
toluene (0.02 M), 60 °C, 12 h

| entry | 1 | R[2]/R[3]/R[4] (2) | yield (ee) of 3 (%) |
|---|---|---|---|
| 1 | **1a** | Me/Bn/Ph (**2a**) | 96 (95, (S)-**3aa**) |
| 2 | **1b** | Me/Bn/Ph (**2a**) | 91 (93, (S)-**3ba**) |
| 3 | **1c** | Me/Bn/Ph (**2a**) | 90 (93, (S)-**3ca**) |
| 4 | **1d** | Me/Bn/Ph (**2a**) | 96 (94, (S)-**3da**) |
| 5 | **1e** | Me/Bn/Ph (**2a**) | >99 (97, (S)-**3ea**) |
| 6[b] | **1f** | Me/Bn/Ph (**2a**) | 86 (78, (R)-**3fa**) |
| 7 | **1a** | Me/o-MeC$_6$H$_5$CH$_2$/Ph (**2b**) | 97 (95, (R)-**3ab**) |
| 8 | **1a** | Me/m-MeC$_6$H$_5$CH$_2$/Ph (**2c**) | 96 (92, (S)-**3ac**) |
| 9 | **1a** | Me/p-MeC$_6$H$_5$CH$_2$/Ph (**2d**) | 92 (95, (S)-**3ad**) |
| 10 | **1a** | Me/p-ClC$_6$H$_5$CH$_2$/Ph (**2e**) | 97 (95, (S)-**3ae**) |
| 11 | **1a** | Me/p-BrC$_6$H$_5$CH$_2$/Ph (**2f**) | 96 (94, (S)-**3af**) |
| 12 | **1a** | Me/p-FC$_6$H$_5$CH$_2$/Ph (**2g**) | 94 (94, (S)-**3ag**) |
| 13 | **1a** | Me/p-NCC$_6$H$_5$CH$_2$/Ph (**2h**) | 97 (94, (S)-**3ah**) |
| 14 | **1a** | Me/p-CF$_3$C$_6$H$_5$CH$_2$/Ph (**2i**) | 96 (94, (S)-**3ai**) |
| 15 | **1a** | Me/m-MeOC$_6$H$_5$CH$_2$/Ph (**2j**) | 95 (94, (S)-**3aj**) |
| 16 | **1a** | Me/m-O$_2$NC$_6$H$_5$CH$_2$/Ph (**2k**) | 86 (98, (S)-**3ak**) |
| 17[c] | **1a** | Me/Et/Ph (**2l**) | 87 (94, (S)-**3al**) |
| 18[d] | **1a** | Me/allyl/Ph (**2m**) | 96 (93, (S)-**3am**) |
| 19[e] | **1a** | Me/allyl/Ph (**2m**) | 95 (94, (R)-**3am**) |
| 20 | **1a** | Me/α-NaphthylCH$_2$/Ph (**2n**) | 95 (97, (R)-**3an**) |
| 21 | **1a** | Me/CH$_2$CH$_2$OH/Ph (**2o**) | 94 (80, (R)-**3ao**) |
| 22 | **1a** | Me/CH$_2$CH$_2$OTBS/Ph (**2p**) | 91 (93, (R)-**3ap**) |
| 23[f,g] | **1a** | Me/Ph/Ph (**2t**) | 62 (36, (R)-**3at**) |
| 24 | **1a** | Et/Bn/Ph (**2q**) | >99 (96, (S)-**3aq**) |
| 25 | **1a** | $^i$Pr/Bn/Ph (**2r**) | 91 (97, (S)-**3ar**) |
| 26 | **1a** | Ph/Bn/Ph (**2s**) | 94 (95, (S)-**3as**) |
| 27[f] | **1b** | Me/Bn/Cy (**2u**) | 84 (95, (S)-**3bu**) |
| 28[f] | **1b** | Me/Bn/Bn (**2v**) | 82 (90, (S)-**3bv**) |

[a]Reaction conditions: **1** (0.5 mmol), **2** (1.2 equiv), Pd$_2$(dba)$_3$•CHCl$_3$ (2.5 mol%), and **DACH-ZYC-Phos-C1** (5.0 mol%) in toluene (25 mL) at 60 °C unless otherwise noted. The ee of **3** was determined by chiral HPLC. [b]The reaction time was 2 h. [c]**2l** (1.4 equiv) was used and the reaction time was 13 h. [d]The reaction time was 11 h. [e]*ent*-**DACH-ZYC-Phos-C1** was used and the reaction time was 11 h. [f]Pd$_2$(dba)$_3$•CHCl$_3$ (3.0 mol%), and **DACH-ZYC-Phos-C1** (6.0 mol%) at 80 °C. [g]Recovery of **1a** was 9% as determined by the [1]H NMR analysis of the crude product using mesitylene as the internal standard. Cbz benzyloxycarbonyl, TMS trimethylsilyl.

pyrazolones on the enantioselectivity could be excluded based on the results of (S)-**3bu** and (S)-**3bv**, we have also tried to analyze the non-covalent interactions (NCI) in **TS_left_Si_a** using the Multiwfn[55,56] and VMD programs[57] and no obvious π–π interactions are found so far from the analysis (see supplementary information for the NCI plot of **TS_left_Si_a**).

The hydrogen bonding in **TS_left_Si_b** features a longer H···N distance of 2.18 Å and a smaller N − H···O bond angle of 163.1°, suggesting a weaker interaction, which estimated to be 4.8 kcal/mol. Similarly, hydrogen-bonding interactions also exist in the structure of **TS_left_Re_a** and **TS_left_Re_b**, which would provide the allenylation product with *R* configuration. **TS_left_Re_a**, is stabilized by the hydrogen bonding formed between the ligand's N-H and the carbonyl oxygen of **2a** anion, which was estimated to be about 6.3 kcal/mol. Moreover, **TS_left_Re_a** suffers the unfavorable interaction between **2a** anion and the methylene-π-allyl moiety, which have been clearly shown in the Newman projection along the forming C–C bond in Fig. 6. Both factors contribute to the lower stability of **TS_left_Re_a** as compared to **TS_left_Si_a** by 4.5 kcal/mol. The bond energy of

the hydrogen bonding in **TS_left_Re_b** was estimated to be about 5.4 kcal/mol, which is around 2.7 kcal/mol weaker than the one in **TS_left_Si_a** (about 8.1 kcal/mol), thus, accounting for the preference of **TS_left_Si_a** over **TS_left_Re_b** by 2.3 kcal/mol. Furthermore, **TS_right_Si** is also stabilized by the hydrogen bonding formed between the ligand's N-H and the carbonyl oxygen of **2a** anion. No corresponding TS is able to be located with hydrogen bonding formed between the ligand's N-H and the hydrazine nitrogen. The bond energy of the hydrogen bonding in **TS_right_Si** was estimated to be about 5.8 kcal/mol. What's more, the steric repulsions between **2a** anion and the methylene-π-allyl moiety destabilize **TS_right_Si**. Both factors lead to the less stability of **TS_right_Si** than **TS_left_Si_a** by 3.3 kcal/mol. In addition, there is no hydrogen bonding existing in **TS_right_Re**. And unfavorable steric interactions between the two phenyl groups of **2a** anion and the **DACH-ZYC-Phos-C1** ligand further destabilize **TS_right_Re**, making it the least stable transition state (10.0 kcal/mol higher in energy than **TS_left_Si_a**). The calculated enantioselectivity, basing on the energy difference between **TS_left_Si_a** and **TS_left_Re_b** of 2.3 kcal/mol, is 94%, which is

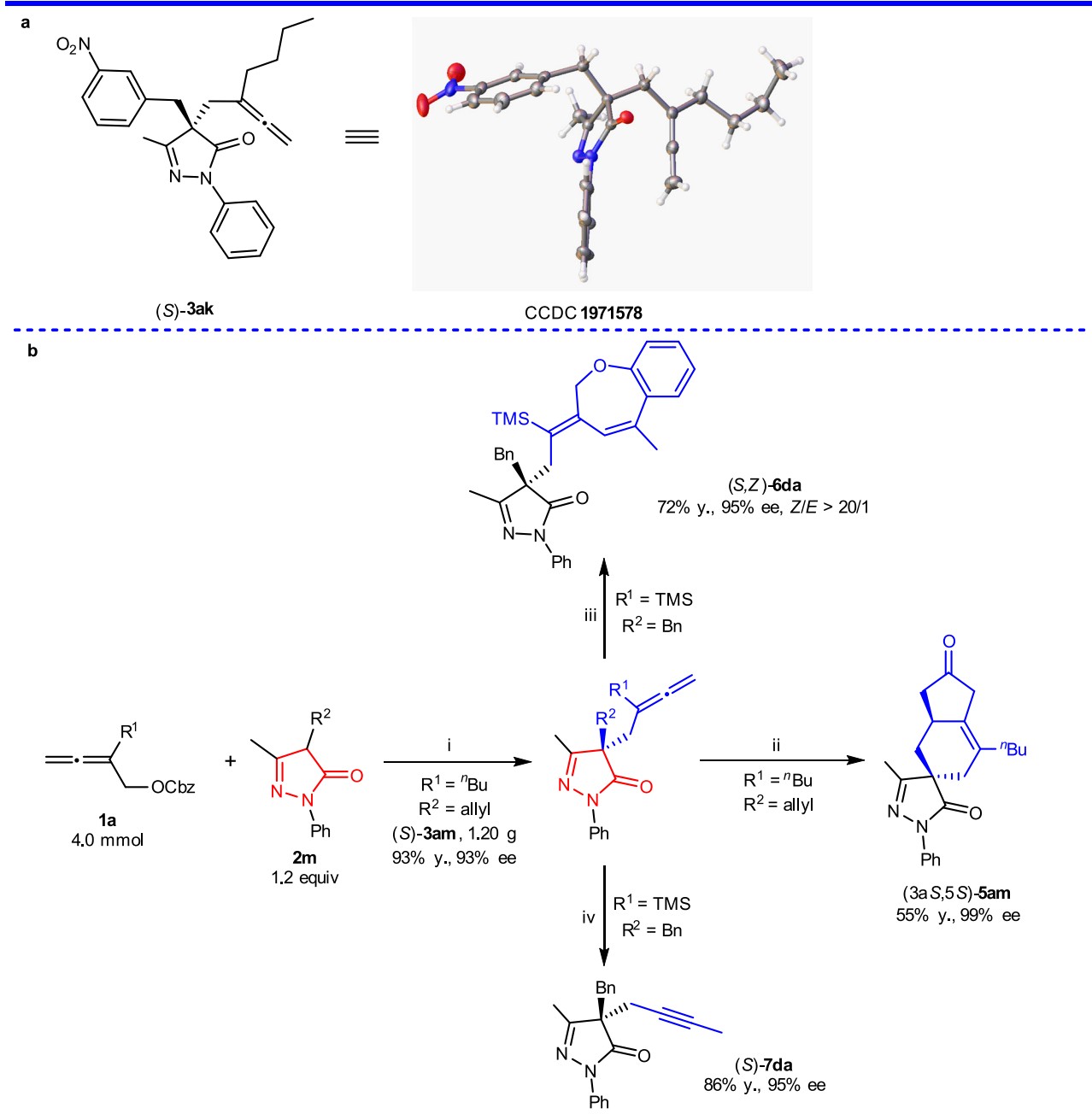

**Fig. 4 ORTEP representation and sythetic applications. a** ORTEP representation of (*S*)-**3ak**. **b** The gram-scale reaction and applications: reaction conditions: (i) Pd₂(dba)₃•CHCl₃ (2.5 mol%), **DACH-ZYC-Phos-C1** (5.0 mol%), toluene, 60 °C, 13 h; (ii) [Rh(CO)₂Cl]₂ (5.0 mol%), AgSbF₆ (12.5 mol%), CO balloon, toluene, 50 °C, 13 h; (iii) ortho-(propen-2-yl)phenol (3.0 equiv), Pd(OAc)₂ (7.5 mol%), 2,2'-bipy (7.5 mol%), Cu(OAc)₂•H₂O (0.5 equiv), toluene, 110 °C, 18 h; (iv) TiCl₄ (1.5 equiv), dichloromethane, −78 °C, 2.5 h, rt, 5.5 h. TMS trimethylsilyl, Cbz benzyloxycarbonyl, y. isolated yield.

in perfect agreement with the experimental ee value of 93% (Table 2, entry 2).

Similar calculations were then conducted to compare with DACH-Phenyl-Trost ligand. The optimized structures and relative free energies of the competing transition states of the enantioselectivity-determining C−C bond formation step, associated with the endo-methylene-π-allyl palladium complexes, are illustrated in Fig. 7a, which are denoted as **TS'_left_Si_a/b**, **TS'_left_Re_a/b** and **TS'_right_Si**, separately. It should be noted that the structure of **TS'_right_Re** has not been located successfully, despite all the efforts, probably due to the severe steric repulsions. Different from the **DACH-ZYC-Phos-C1** participated reaction, all the transition structures with methylene moiety lying

on the left side to the π-allyl Pd unit suffer the steric interactions causing by **2a** anion with the phenyl group in phenyl-Trost ligand, which reduce their stability. Nevertheless, via avoiding the unfavorable hindrance caused by the nucleophile **2a** anion with the phenyl group in phenyl-Trost ligand, **TS'_right_Si** becomes the most favorable transition state. The energy difference between the diastereomeric **TS'_right_Si** and **TS'_left_Re_b** is found to be 1.2 kcal/mol. The calculated enantioselectivity of 71% in favor of also the *S* enantiomer is in agreement with the experimental ee value of 71% (Fig. 7b).

In summary, we have developed a DPPAA-based ligand for the highly chemo- and enantioselective allenylation of pyrazol-5-ones with benzyl 2,3-dienyl carbonates (up to 97% ee). Many

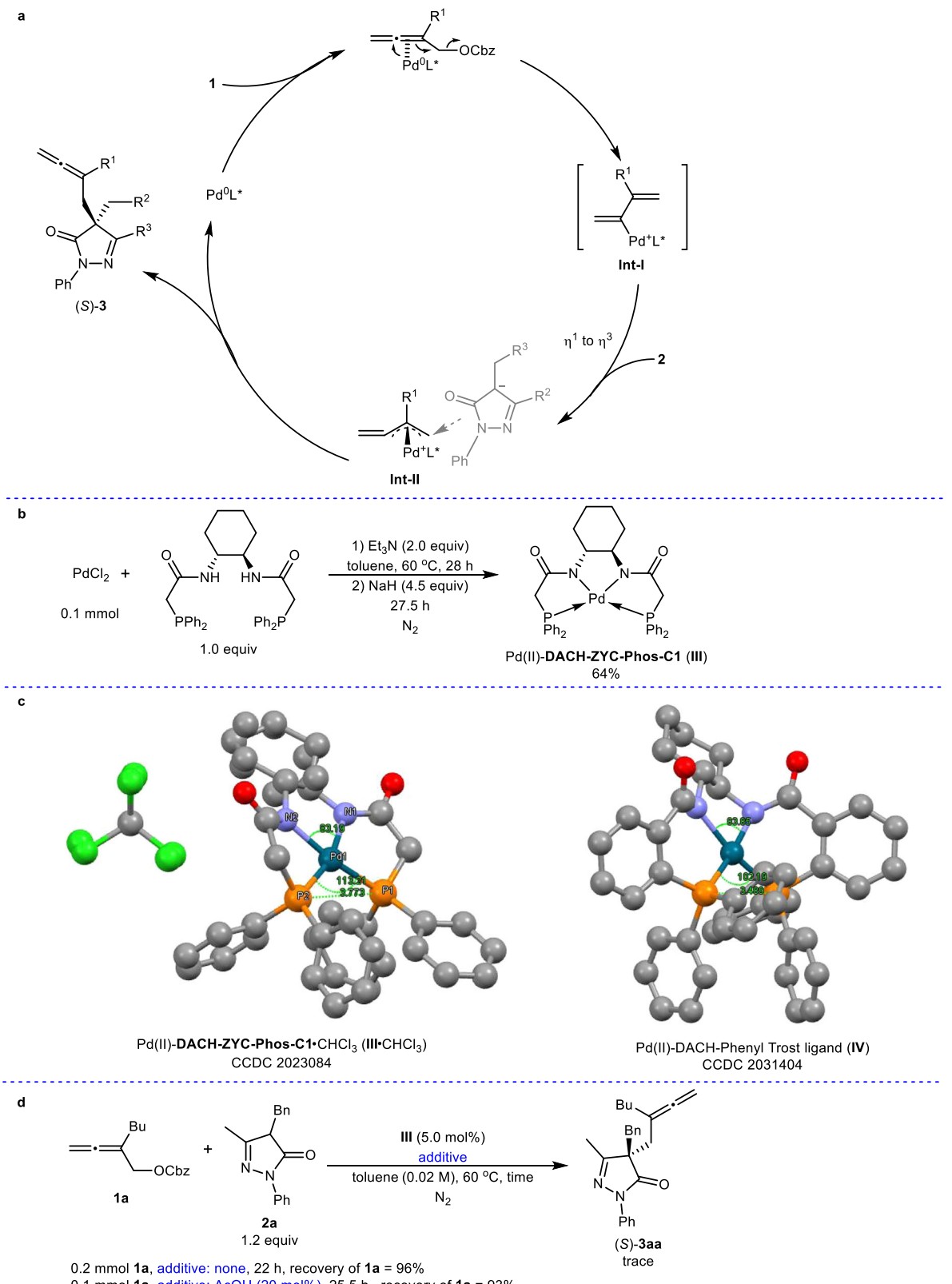

**Fig. 5 A proposal mechanism and some experiments for Pd-ligand complexes. a** A mechanism for the allenylation of pyrazolone. **b** Synthesis of Pd(II)-**DACH-ZYC-Phos-C1** complex **III**. **c** ORTEP representations of Pd(II)-ligand complexes **III** and **IV**. H-atoms are omitted for clarity. **d** The reaction of pyrazolone **2a** and allene **1a** with complex **III**. Cbz benzyloxycarbonyl, DACH (1*R*,2*R*)-1,2-diaminocyclohexane.

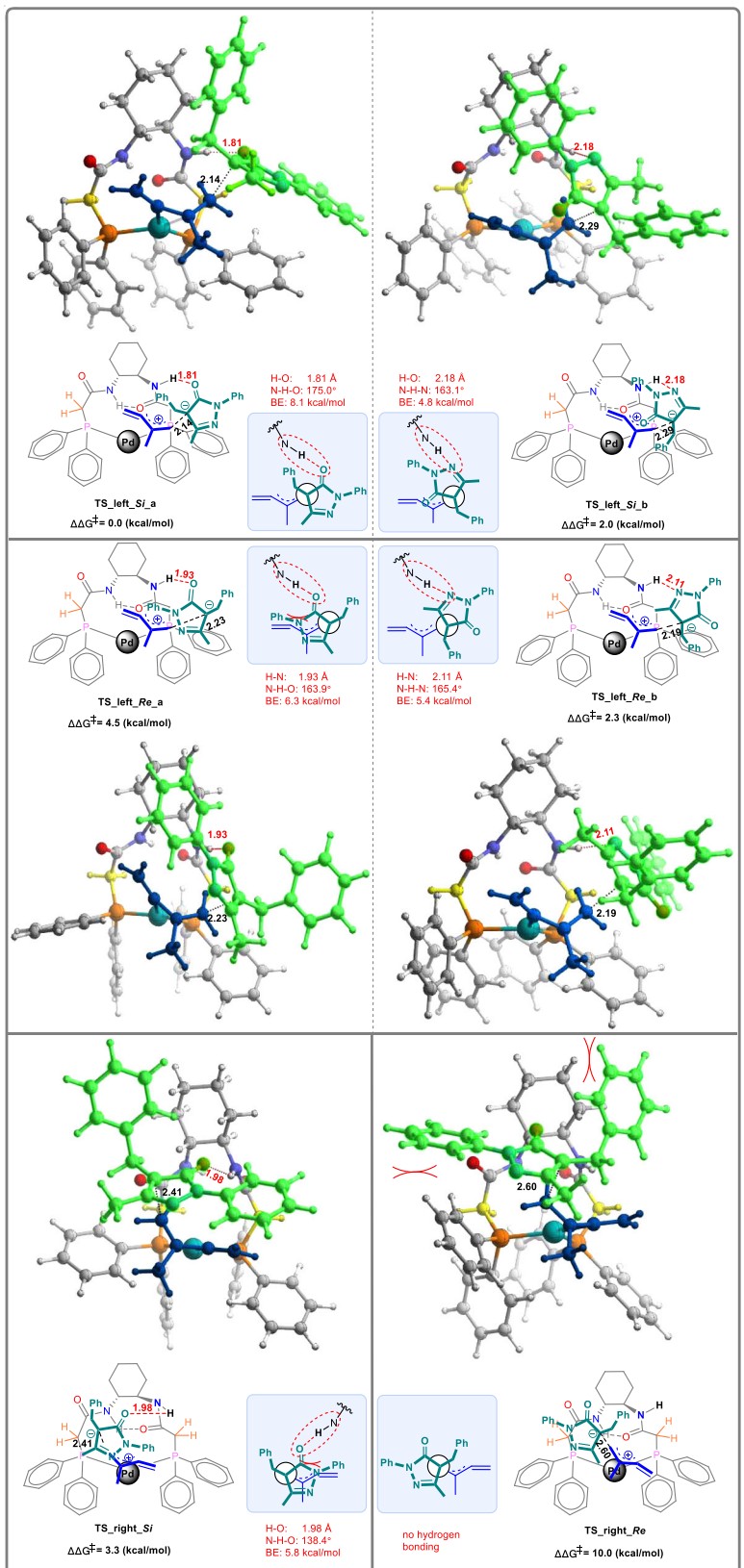

**Fig. 6 DFT calculations for DACH-ZYC-Phos-C1-ligated palladium catalyst.** DFT-optimized structures and relative energies of stereo-determining C−C bond formation transition states of the reaction of **1b** with **2a** catalyzed by the **DACH-ZYC-Phos-C1**-ligated palladium catalyst. Hydrogen-bonding interactions are shown as red dash lines; selected bond lengths (Å) and bond angles (°) are listed. BE bond energy.

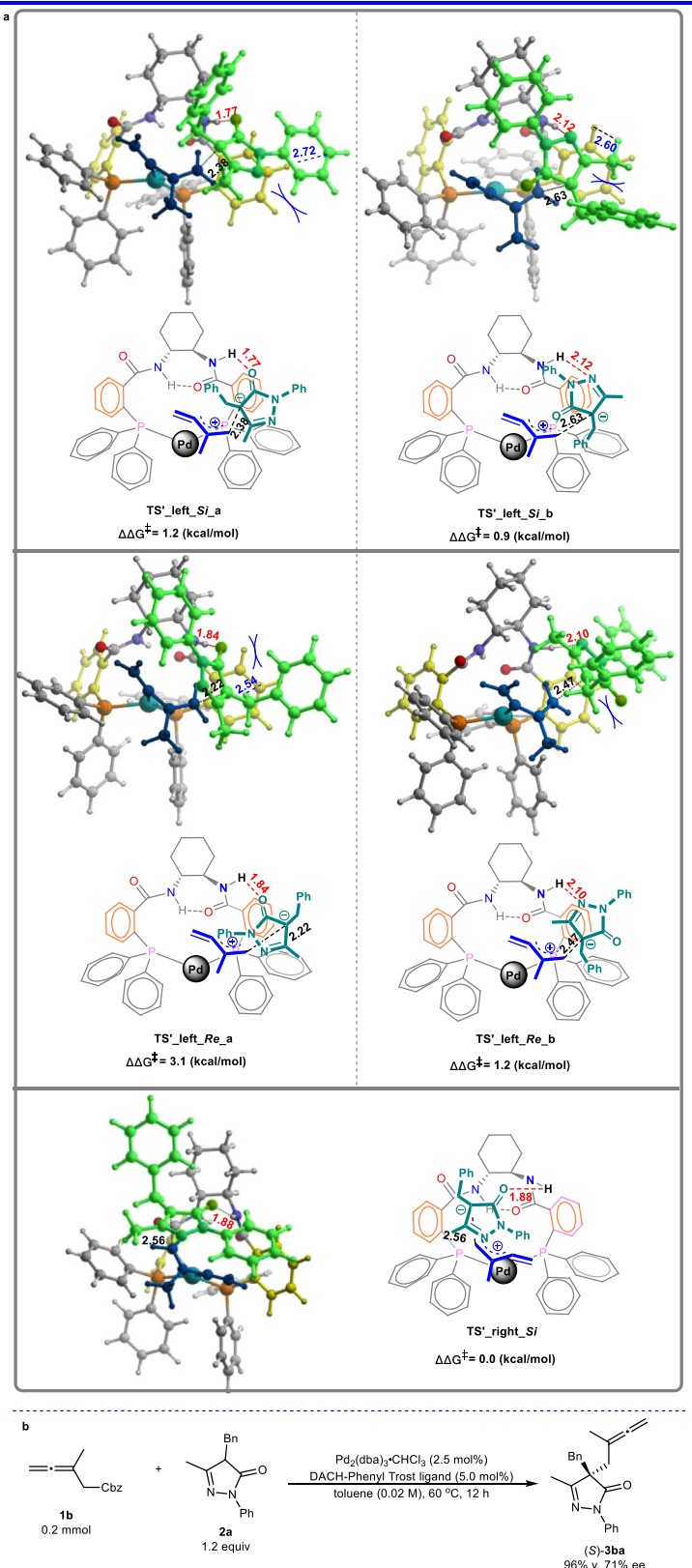

**Fig. 7 DFT calculations for the DACH-Phenyl-Trost-ligated palladium catalyst. a** DFT-optimized structures and relative energies of ee determining C−C bond formation transition states for the reaction of **1b** with **2a** catalyzed by the DACH-Phenyl-Trost-ligated palladium catalyst. Hydrogen-bonding interactions are shown as red dash lines; selected bond lengths (Å) are listed. **b** The reaction of **1b** and **2a** with DACH-Phenyl-Trost ligand. Cbz benzyloxycarbonyl, DACH (1 *R*,2 *R*)-1,2-diaminocyclohexane, y. NMR yield.

synthetically useful functionalities were tolerated under the catalysis of Pd/**DACH-ZYC-Phos-C1**. A rationale has been provided based on the X-ray diffraction studies of Pd(II)-**DACH-ZYC-Phos-C1** complex and DFT calculations. In addition, these types of stretchable chiral pockets may also provide flexiblity and show great potential applications in catalytic enantioselective allylation, allenylation, and other reactions. Such studies are being actively pursued in our laboratory with very promising results and will be reported in due courses.

## Methods

**General procedure for the catalytic enantioselective allenylation of pyrazol-5-ones.** To a flame-dried Schlenk flask were added Pd$_2$(dba)$_3$•CHCl$_3$ (12.9 mg, 0.0125 mmol), (*R,R*)-**DACH-ZYC-Phos-C1** (14.3 mg, 0.025 mmol), toluene (5 mL), **1a** (131.5 mg, 0.5 mmol)/toluene (11.7 mL), and **2a** (158.7 mg, 0.6 mmol)/toluene (8.3 mL) sequentially under nitrogen atmosphere at room temperature. The flask was put into a pre-heated oil bath and the reaction was complete after being stirred at 60 °C for 12 h as monitored by TLC (eluent: petroleum ether/ethyl acetate = 20/1). The resulting mixture was filtrated through a short column of silica gel eluted with ethyl acetate (15 mL × 3). After evaporation, the crude residue was purified by chromatography on silica gel (eluent: petroleum ether (60–90 °C)/ethyl acetate = 60/1) to afford a pure part of (*S*)-**3aa** and the impure part was further purified by chromatography on silica gel (eluent: petroleum ether (60–90 °C)/ethyl acetate = 40/1). Two-round chromatography afforded (*S*)-**3aa** (179.5 mg, 96%) as a liquid: 95% ee (HPLC condition: Chiralcel IA column, *n*-hexane/*i*-PrOH = 90/10, 1.0 mL/min, $\lambda$ = 254 nm, $t_R$ (major) = 5.7 min, $t_R$ (minor) = 8.9 min); [$\alpha$]$_D^{20}$ = −8.2 (*c* = 1.25, CHCl$_3$); $^1$H NMR (300 MHz, CDCl$_3$) δ 7.55 (*d*, *J* = 8.1 Hz, 2 H, ArH), 7.30 (*t*, *J* = 7.8 Hz, 2 H, ArH), 7.22–7.02 (m, 6 H, ArH), 4.64–4.49 (m, 2 H, = CH$_2$), 3.18 (*d*, *J* = 13.2 Hz, 1 H, one proton of CH$_2$), 2.89 (*d*, *J* = 13.5 Hz, 1 H, one proton of CH$_2$), 2.69 (dt, $J_1$ = 15.3 Hz, $J_2$ = 3.3 Hz, 1 H, one proton of CH$_2$), 2.39 (*d*, *J* = 15.0 Hz, 1 H, one proton of CH$_2$), 2.14 (s, 3 H, CH$_3$), 1.93–1.80 (m, 2 H, CH$_2$), 1.43–1.17 (m, 4 H, CH$_2$ × 2), 0.83 (*t*, *J* = 7.1 Hz, 3 H, CH$_3$); $^{13}$C NMR (75 MHz, CDCl$_3$) δ 205.3, 174.6, 161.4, 137.6, 134.1, 129.1, 128.5, 128.1, 127.2, 124.8, 119.3, 98.3, 77.7, 60.1, 42.8, 36.1, 32.3, 29.4, 22.1, 14.7, 13.8; IR (neat) ν (cm$^{-1}$) 3063, 3031, 2956, 2927, 2859, 1954, 1712, 1597, 1500, 1455, 1440, 1402, 1366, 1123; MS (EI): *m/z* (%) 372 ([M]$^+$, 31.27), 186 (100); HRMS calcd. for C$_{25}$H$_{28}$N$_2$O [M]$^+$: 372.2202; Found: 372.2202.

## Data availability

All data that support the findings of this study are available in the online version of this paper in the accompanying Supplementary Information (including experimental procedures, compound characterization data, and spectra). The X-ray crystallographic coordinates for structures of (*S*)-**3ak**, Pd(II)-**DACH-ZYC-Phos-C1** complex and Pd(II)-DACH-Phenyl Trost ligand complex reported in this article have been deposited at the Cambridge Crystallographic Data Centre (CCDC) under deposition numbers CCDC 1971578 ((*S*)-**3ak**), 2023084 (Pd(II)-**DACH-ZYC-Phos-C1** complex) and 2031404 (Pd (II)-DACH-Phenyl Trost ligand complex). These data can be obtained free of charge from http://www.ccdc.cam.ac.uk/data_request/cif.

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

## Acknowledgements

Financial support from the National Natural Science Foundation of China (Grant Nos. 21690063 and 21988101) is greatly appreciated. We thank Haibo Xu in this group for reproducing the preparation of (*S*)-**3ba**, (*S*)-**3af**, and (*S*)-**3ar**. Shengming Ma is a Qiu Shi Adjunct Professor at Zhejiang University.

## Author contributions

S.M. directed the research and developed the concept of the reaction with Y. Z., who also performed the experiments and prepared the Supplementary Information. X.Z. performed the computational studies. Y.Z., X.Z., and S.M. wrote the manuscript and checked the experimental data.

## Competing interests

The authors declare no competing interests.
