## [Peer Review File · Nature Communications]

REVIEWER COMMENTS

Reviewer #1 (Remarks to the Author):

In this manuscript submitted for publication in Nature Communications, the authors describe a palladium-catalyzed enantioselective allenylation of pyrazol-5-ones with 2,3-allenyl carbonates. The reaction is achieved with high yields and regio- and enantioselectivities by using a novel DACH-type bidentate phosphine ligand, which is modified from a long-known Trost style DACH ligands. It is worth noting that minor modifications of the flexibility of the ligand result in much better enantioselectivities. Particularly, DFT calculations provide deep insight into the origin of the enantioselectivity and also the superiority of the new ligand to Trost ligand. The manuscript was carefully prepared and is well written. Therefore, I strongly recommend its acceptance in Nature Communications after minor revision.

1. According to the results shows in Table 2, DAHC-ZYC-Phos-C1 seems to be superior to other ligands with longer alkyl linker in controlling the enantioselectivity. However, I noticed that DAHC-ZYC-Phos-C2 also gives similar yield and enantioselectivity in CHCl₃. I suggest that the authors should further test the DAHC-ZYC-Phos-C1/C2/C3 ligands in the standard condition (in toluene).
2. In the X-ray single crystal (scheme 4c), the bite angle (P-Pd-P) and the distance between two P atoms in III are greater than those in IV. I suggest that the authors should also check these data in the corresponding transition structures of the DFT simulation, which might help to further understand the mechanism of the reaction.

Reviewer #2 (Remarks to the Author):

This manuscript by Ma describes an asymmetric allenylation of pyrazol-5-ones. The authors have previously disclosed the enantioselective spirocyclizations of pyrazolones with 2,3-allenyl acetates. So the major observation in this paper is an interesting exploration and extension of Trost phosphine ligands from rigid aryl to flexible alkyl linker with improvement on stereocontrol. Some synthetically useful transformations were demonstrated. I'd like to recommend publication after the following revisions:

- 1) The author stated 'the complex III is not really a catalytically active species but provides a stable coordination mode', and they moved to calculate on this structure with N-H moiety presented. There appears a missing link between experimental observation and the DFT calculation. Is it possible this inactive complex being activated by adding protic additive? It's known that in Trost-allylation with the similar ligand, the N-H deprotonation status is critical. Have the authors considered the mono-deprotonated status for catalysis?
- 2) Table 3, entries 6 and 19-21, please specify why the configurations are R instead of S- as observed in other cases under the optimized conditions.
- 3) Line 110, please delete "in addition to"
- 4) What about unsubstituted allenes (R1 = H)?

Reviewer #3 (Remarks to the Author):

This beautiful manuscript is a complete study to development a new enantioselective methodology to construct pyrazol-5-ones skeletons. The authors, clearly and concisely, describe the design of the new methodology, not only the optimization of the reaction conditions and the best ligand, but also the synthesis of a new ligand, designed ad-hoc for this process, and the DFT study that explains the results obtained.

The characterization and purity of the products is high quality. However, to improve the understanding of the results obtained, it is recommended to take into account several considerations:

1. the authors use pyrazolones substituted in position 4 by an alkyl group and N-phenyl-pyrazolones. What happens when these positions are replaced by other groups different from these? The authors should indicate if it has been taken into account in the scope or not.
2. The results of the DFT study should be shown more clearly: a. Firstly, discuss which heteroatom forms the hydrogen bond with N-H, that is, oxygen versus nitrogen; b. then and established the hydrogen bond between the oxygen of the pyrazolone and the N-H of the ligand, discuss the four possibilities of the transition state; c. For clarity, the descriptors should indicate the face of each of the components that reacts, that is, there would be 4 possible transition states to show and analyze: TS_{Re-Re}, TS_{Re-Si}, TS_{Si-Si} and TS_{Si-Re}
3. In addition to the steric interactions and the importance of the hydrogen bond, the authors should show the distances of the bonds formed in transition states.
4. Taking into account that the authors use N-Ph pyrazolones, it would be of interest to analyze the contribution of the pi-pi interactions.
5. It is recommended not to use the descriptors endo and exo
6. the manuscript must be thoroughly revised to avoid typographical errors

Few precedents exist in the literature for the stereoselective functionalization of pyrazolones

Reviewers' Comments to Author:

Reviewer #1 (Remarks to the Author):

In this manuscript submitted for publication in Nature Communications, the authors describe a palladium-catalyzed enantioselective allenylation of pyrazol-5-ones with 2,3-allenyl carbonates. The reaction is achieved with high yields and regio- and enantioselectivities by using a novel DACH-type bidentate phosphine ligand, which is modified from a long-known Trost style DACH ligands. It is worth noting that minor modifications of the flexibility of the ligand result in much better enantioselectivities. Particularly, DFT calculations provide deep insight into the origin of the enantioselectivity and also the superiority of the new ligand to Trost ligand. The manuscript was carefully prepared and is well written. Therefore, I strongly recommend its acceptance in Nature Communications after minor revision.

1. According to the results shows in Table 2, DACH-ZYC-Phos-C1 seems to be superior to other ligands with longer alkyl linker in controlling the enantioselectivity. However, I noticed that DACH-ZYC-Phos-C2 also gives similar yield and enantioselectivity in CHCl₃. I suggest that the authors should further test the DACH-ZYC-Phos-C1/C2/C3 ligands in the standard condition (in toluene).

Response: Many thanks for suggestions. Using toluene as solvent, **DACH-ZYC-Phos-C2** and **C3** were tested and the results are added to entries 14 and 15 of Table 2. **DACH-ZYC-Phos-C1** is still the best.

Table 2. The effect of temperature, concentration, ligands and solvents on the asymmetric allenylation of pyrazolone **2a** with allene **1a**.^a

entry	ZYC-Phos	solvent (v)	yield (ee) of (S)-3aa/4aa/1a recovered (%) ^b
1 ^c	DACH-ZYC-Phos-C2	CHCl_3 (3)	14 (85)/trace/67
2	DACH-ZYC-Phos-C2	CHCl_3 (3)	75 (87)/4/6
3 ^d	DACH-ZYC-Phos-C2	CHCl_3 (3)	80 (86)/4/0
4 ^e	DACH-ZYC-Phos-C2	CHCl_3 (1.5)	67 (82)/7/4
5 ^e	DACH-ZYC-Phos-C2	CHCl_3 (5)	78 (87)/3/4
6 ^f	DACH-ZYC-Phos-C2	CHCl_3 (10)	70 (88)/2/12
7	DACH-ZYC-Phos-C2	CHCl_3 (10)	76 (89)/2/17
8	DADPE-ZYC-Phos-C2	CHCl_3 (10)	79 (79)/5/8
9 ^g	DACH-ZYC-Phos-C3	CHCl_3 (10)	84 (76)/7/0
10	DACH-ZYC-Phos-C1	CHCl_3 (10)	79 (90)/2/10
11 ^h	DACH-ZYC-Phos-C1	toluene (10)	100 (95)/0/0
12	DACH-ZYC-Phos-C1	THF (10)	92 (83)/3/0
13	DACH-ZYC-Phos-C1	CH_3CN (10)	23 (n.d.)/4/60
14	DACH-ZYC-Phos-C2	toluene (10)	87 (96)/0/11
15	DACH-ZYC-Phos-C3	toluene (10)	90 (87)/0/0

^a Reaction conditions: **1a** (0.2 mmol), **2a** (1.2 equiv), $\text{Pd}_2(\text{dba})_3 \cdot \text{CHCl}_3$ (2.5 mol%), and ligand (6 mol% for entries 1-6; 5 mol% for entries 7-13) unless otherwise noted. ^b The yields of **(S)-3aa** and **4aa** as well as the recovery of **1a** were determined by the ¹H NMR analysis of the crude product using mesitylene as the internal standard and the ee of isolated **(S)-3aa** was determined by chiral HPLC. ^c At 30 °C for 20 h. ^d At 80 °C. ^e The reaction time was 14 h. ^f The reaction time was 16 h. ^g The reaction time was 12.5 h. ^h The reaction was conducted on 0.5 mmol scale.

2. In the X-ray single crystal (scheme 4c), the bite angle (P-Pd-P) and the distance between two P atoms in III are greater than those in IV. I suggest that the authors should also check these data in the corresponding transition structures of the DFT simulation, which might help to further understand the mechanism of the reaction.

Response: Thanks for your suggestion. As you suggested, we have checked the bite angles ($\text{P}^1\text{-Pd-P}^2$) and the distances between two P atoms ($\text{P}^1\text{-P}^2$) in the transition structures. The comparisons of the data are listed as follows. No obvious trends are found from the table. We considered that the transition structures are relatively flexible than the corresponding X-Ray structures, thus, the bite angles and the distances of $\text{P}^1\text{-P}^2$ in the TS could be adjusted to make the structures more stable.

		P ¹ -Pd-P ² (°)	P ¹ -P ² (Å)
Pd(II)-DACH-ZYC-Phos-C1 ligand	complex III	113.21	3.773
	TS_left_Si_a	111.23	3.927
	TS_left_Si_b	111.41	3.914
	TS_left_Re_a	107.18	3.821
	TS_left_Re_b	110.85	3.911
	TS_right_Si	104.43	3.705
	TS_right_Re	103.78	3.701
Pd(II)-DACH-Phenyl Trost ligand	complex IV	102.19	3.489
	TS'_left_Si_a	111.49	3.975
	TS'_left_Si_b	113.84	4.031
	TS'_left_Re_a	113.80	4.012
	TS'_left_Re_b	111.94	3.985
	TS'_right_Si	110.67	3.937

Reviewer #2 (Remarks to the Author):

This manuscript by Ma describes an asymmetric allenylation of pyrazol-5-ones. The authors have previously disclosed the enantioselective spirocyclizations of pyrazolones with 2,3-allenyl acetates. So the major observation in this paper is an interesting exploration and extension of Trost phosphine ligands from rigid aryl to flexible alkyl linker with improvement on stereocontrol. Some synthetically useful transformations were demonstrated. I'd like to recommend publication after the following revisions:

1) The author stated ' the complex **III** is not really a catalytically active species but provides a stable coordination mode', and they moved to calculate on this structure with N-H moiety presented. There appears a missing link between experimental observation and the DFT calculation. Is it possible this inactive complex being activated by adding protic additive? It's known that in Trost-allylation with the similar ligand, the N-H deprotonation status is critical. Have the authors considered the mono-deprotonated status for catalysis?

Response: Thanks for your comments. We have used AcOH as a protic additive, but no products were obtained with 93% recovery of **1a**. This indicated that protonation of complex **III** to form the catalytically active species is NOT possible. This result has been added to Scheme 4d with the following description: "However, this complex **III** failed to catalyze the enantioselective allenylation under the standard conditions or loading AcOH as protic additive: no product (*S*)-**3fa** was obtained with 96% or 93% recovery of **1a** (Scheme 4d)."

Scheme 4. a, A mechanism for the allenylation of pyrazolone. **b**, Synthesis of Pd(II)-DACH-ZYC-Phos-C1 complex **III**. **c**, ORTEP representations of Pd(II)-ligand complexes **III** and **IV**. H-atoms are omitted for clarity. **d**, The reaction of pyrazolone **2a** and allene **1a** with complex **III**.

Furthermore, we thought one of the reactants of this reaction is methylene- π -allyl, which acts as a η^2 -ligand to the palladium center. Thus, we considered that the **DACH-ZYC-Phos-C1** and DACH-

Phenyl Trost ligand both acted as diphosphine ligands in the reaction system.

2) Table 3, entries 6 and 19-21, please specify why the configurations are R instead of S- as observed in other cases under the optimized conditions.

Response: The absolute configurations of the products are same. It is the change of priority between the allene group and the *o*-methylbenzenyl, α -naphthylmethyl, 2-hydroxyethyl, 2-((tert-butyl)dimethylsilyloxy)ethyl at 4-position that leads to the R-configurations in Table 3, entries 7 and 20-22 (corresponding to Table 3, entries 6 and 19-21 in last edition of the manuscript). It is just an issue of nomenclature.

3) Line 110, please delete “in addition to”

Response: We have revised the sentence “For R¹, in addition to *n*-butyl, methyl, *n*-octyl, TMS, and phenyl all worked well with an excellent yield and ee (Table 3, entries 1-5)” to “For R¹, *n*-butyl, methyl, *n*-octyl, TMS, phenyl, and H all worked well with an excellent yield and ee (Table 3, entries 1-6)”.

4) What about unsubstituted allenes (R¹ = H)?

Response: We have conducted the reaction of this suggested substrate and added the result as entry 6 in Table 3 with the revision to the following text: “However, when R¹ is H, the ee value of (*R*)-**3fa** dropped to 78% with 86% yield, indicating a steric effect (Table 4, entry 6).”

Table 3. The Scope of allenes and pyrazol-5-ones. ^a
entry	1	$\text{R}^2/\text{R}^3/\text{R}^4$ (2)	yield (ee) of 3 (%)
1	1a	Me/Bn/Ph (2a)	96 (95, (S)- 3aa)
2	1b	Me/Bn/Ph (2a)	91 (93, (S)- 3ba)
3	1c	Me/Bn/Ph (2a)	90 (93, (S)- 3ca)
4	1d	Me/Bn/Ph (2a)	96 (94, (S)- 3da)
5	1e	Me/Bn/Ph (2a)	>99 (97, (S)- 3ea)
6^b	1f	Me/Bn/Ph (2a)	86 (78, (R)-3fa)
7	1a	Me/ o -MeC ₆ H ₅ CH ₂ /Ph (2b)	97 (95, (R)- 3ab)
8	1a	Me/ m -MeC ₆ H ₅ CH ₂ /Ph (2c)	96 (92, (S)- 3ac)
9	1a	Me/ p -MeC ₆ H ₅ CH ₂ /Ph (2d)	92 (95, (S)- 3ad)
10	1a	Me/ p -ClC ₆ H ₅ CH ₂ /Ph (2e)	97 (95, (S)- 3ae)
11	1a	Me/ p -BrC ₆ H ₅ CH ₂ /Ph (2f)	96 (94, (S)- 3af)
12	1a	Me/ p -FC ₆ H ₅ CH ₂ /Ph (2g)	94 (94, (S)- 3ag)
13	1a	Me/ p -NCC ₆ H ₅ CH ₂ /Ph (2h)	97 (94, (S)- 3ah)
14	1a	Me/ p -CF ₃ C ₆ H ₅ CH ₂ /Ph (2i)	96 (94, (S)- 3ai)
15	1a	Me/ m -MeOC ₆ H ₅ CH ₂ /Ph (2j)	95 (94, (S)- 3aj)
16	1a	Me/ m -O ₂ NC ₆ H ₅ CH ₂ /Ph (2k)	86 (98, (S)- 3ak)
17 ^c	1a	Me/Et/Ph (2l)	87 (94, (S)- 3al)
18 ^d	1a	Me/allyl/Ph (2m)	96 (93, (S)- 3am)
19 ^e	1a	Me/allyl/Ph (2m)	95 (94, (R)- 3am)
20	1a	Me/ α -NaphthylCH ₂ /Ph (2n)	95 (97, (R)- 3an)
21	1a	Me/CH ₂ CH ₂ OH/Ph (2o)	94 (80, (R)- 3ao)
22	1a	Me/CH ₂ CH ₂ OTBS/Ph (2p)	91 (93, (R)- 3ap)
23 ^{f,g}	1a	Me/Ph/Ph (2t)	62 (36, (R)- 3at)
24	1a	Et/Bn/Ph (2q)	>99 (96, (S)- 3aq)
25	1a	ⁱ Pr/Bn/Ph (2r)	91 (97, (S)- 3ar)
26	1a	Ph/Bn/Ph (2s)	94 (95, (S)- 3as)
27 ^f	1b	Me/Bn/Cy (2u)	84 (95, (S)- 3bu)
28 ^f	1b	Me/Bn/Bn (2v)	82 (90, (S)- 3bv)

^a Reaction conditions: **1** (0.5 mmol), **2** (1.2 equiv), Pd₂(dba)₃•CHCl₃ (2.5 mol%), and **DAHC-ZYC-Phos-C1** (5.0 mol%) in toluene (25 mL) at 60 °C unless otherwise noted. The ee of **3** was determined by chiral HPLC. ^b The reaction time was 2 h. ^c **2l** (1.4 equiv) was used and the reaction time was 13 h. ^d The reaction time was 11 h. ^e *ent*-**DAHC-ZYC-Phos-C1** was used and the reaction time was 11 h. ^f Pd₂(dba)₃•CHCl₃ (3.0 mol%), and **DAHC-ZYC-Phos-C1** (6.0 mol%) at 80 °C. ^g Recovery of **1a** was 9% as determined by the ¹H NMR analysis of the crude product using mesitylene as the internal standard.

Reviewer #3 (Remarks to the Author):

This beautiful manuscript is a complete study to develop a new enantioselective methodology to construct pyrazol-5-ones skeletons. The authors, clearly and concisely, describe the design of the new methodology, not only the optimization of the reaction conditions and the best ligand, but also the synthesis of a new ligand, designed ad-hoc for this process, and the DFT study that explains the results obtained.

The characterization and purity of the products is high quality. However, to improve the understanding of the results obtained, it is recommended to take into account several considerations: 1. the authors use pyrazolones substituted in position 4 by an alkyl group and N-phenyl-pyrazolones. What happens when these positions are replaced by other groups different from these? The authors should indicate if it has been taken into account in the scope or not.

Response: Many thanks for the suggestions. We have conducted the reactions of 4-phenyl, N-cyclohexyl and N-benzyl-pyrazolones and added the results as entries 23, 27, and 28 in Table 3 with the following descriptions: “(1) Furthermore, for R³ is phenyl, the ee value of (*R*)-**3at** was only 36% with 62% yield (Table 3, entry 23). (2) The reaction of *N*-cyclohexyl pyrazol-5-one **2u** and **1b** afforded (*S*)-**3bu** in 84% yield with 95% ee, while the reaction of *N*-benzyl pyrazol-5-one **2v** and **1b** afforded (*S*)-**3bv** in 82% yield with 90% ee (Table 3, entries 27 and 28).”

Table 3. The Scope of allenes and pyrazol-5-ones. ^a
entry	1	$\text{R}^2/\text{R}^3/\text{R}^4$ (2)	yield (ee) of 3 (%)
1	1a	Me/Bn/Ph (2a)	96 (95, (S)- 3aa)
2	1b	Me/Bn/Ph (2a)	91 (93, (S)- 3ba)
3	1c	Me/Bn/Ph (2a)	90 (93, (S)- 3ca)
4	1d	Me/Bn/Ph (2a)	96 (94, (S)- 3da)
5	1e	Me/Bn/Ph (2a)	>99 (97, (S)- 3ea)
6 ^b	1f	Me/Bn/Ph (2a)	86 (78, (R)- 3fa)
7	1a	Me/ o -MeC ₆ H ₅ CH ₂ /Ph (2b)	97 (95, (R)- 3ab)
8	1a	Me/ m -MeC ₆ H ₅ CH ₂ /Ph (2c)	96 (92, (S)- 3ac)
9	1a	Me/ p -MeC ₆ H ₅ CH ₂ /Ph (2d)	92 (95, (S)- 3ad)
10	1a	Me/ p -ClC ₆ H ₅ CH ₂ /Ph (2e)	97 (95, (S)- 3ae)
11	1a	Me/ p -BrC ₆ H ₅ CH ₂ /Ph (2f)	96 (94, (S)- 3af)
12	1a	Me/ p -FC ₆ H ₅ CH ₂ /Ph (2g)	94 (94, (S)- 3ag)
13	1a	Me/ p -NCC ₆ H ₅ CH ₂ /Ph (2h)	97 (94, (S)- 3ah)
14	1a	Me/ p -CF ₃ C ₆ H ₅ CH ₂ /Ph (2i)	96 (94, (S)- 3ai)
15	1a	Me/ m -MeOC ₆ H ₅ CH ₂ /Ph (2j)	95 (94, (S)- 3aj)
16	1a	Me/ m -O ₂ NC ₆ H ₅ CH ₂ /Ph (2k)	86 (98, (S)- 3ak)
17 ^c	1a	Me/Et/Ph (2l)	87 (94, (S)- 3al)
18 ^d	1a	Me/allyl/Ph (2m)	96 (93, (S)- 3am)
19 ^e	1a	Me/allyl/Ph (2m)	95 (94, (R)- 3am)
20	1a	Me/ α -NaphthylCH ₂ /Ph (2n)	95 (97, (R)- 3an)
21	1a	Me/CH ₂ CH ₂ OH/Ph (2o)	94 (80, (R)- 3ao)
22	1a	Me/CH ₂ CH ₂ OTBS/Ph (2p)	91 (93, (R)- 3ap)
23^{f,g}	1a	Me/Ph/Ph (2t)	62 (36, (R)-3at)
24	1a	Et/Bn/Ph (2q)	>99 (96, (S)- 3aq)
25	1a	ⁱ Pr/Bn/Ph (2r)	91 (97, (S)- 3ar)
26	1a	Ph/Bn/Ph (2s)	94 (95, (S)- 3as)
27^f	1b	Me/Bn/Cy (2u)	84 (95, (S)-3bu)
28^f	1b	Me/Bn/Bn (2v)	82 (90, (S)-3bv)

^a Reaction conditions: **1** (0.5 mmol), **2** (1.2 equiv), Pd₂(dba)₃•CHCl₃ (2.5 mol%), and **DAHC-ZYC-Phos-C1** (5.0 mol%) in toluene (25 mL) at 60 °C unless otherwise noted. The ee of **3** was determined by chiral HPLC. ^b The reaction time was 2 h. ^c **2l** (1.4 equiv) was used and the reaction time was 13 h. ^d The reaction time was 11 h. ^e *ent*-**DAHC-ZYC-Phos-C1** was used and the reaction time was 11 h. ^f Pd₂(dba)₃•CHCl₃ (3.0 mol%), and **DAHC-ZYC-Phos-C1** (6.0 mol%) at 80 °C. ^g Recovery of **1a** was 9% as determined by the ¹H NMR analysis of the crude product using mesitylene as the internal standard.

2. The results of the DFT study should be shown more clearly: a. Firstly, discuss which heteroatom forms the hydrogen bond with N-H, that is, oxygen versus nitrogen; b. then and established the hydrogen bond between the oxygen of the pyrazolone and the N-H of the ligand, discuss the four possibilities of the transition state; c. For clarity, the descriptors should indicate the face of each of the components that reacts, that is, there would be 4 possible transition states to show and analyze: TS_{Re-Re}, TS_{Re-Si}, TS_{Si-Si} and TS_{Si-Re}.

Response: Thank you for your suggestions.

For issues a and b: Actually, **TS_{left_Si}** and **TS_{left_Re}** each has two conformations, which associated with the hydrogen bonding of the ligand's N-H with the carbonyl oxygen (**a**) or the hydrazine nitrogen (**b**) of **2a** anion. However, for **TS_{right_Si}**, no TS is able to be located with hydrogen bonding formed between the ligand's N-H and the hydrazine nitrogen. And there is no hydrogen bonding existing in **TS_{right_Re}**. Thus, there are totally six transition structures for the **DACH-ZYC-Phos-C1** system, which are denoted as **TS_{left_Si_a/b}**, **TS_{left_Re_a/b}**, **TS_{right_Si}** and **TS_{right_Re}**, separately. Accordingly, for the DACH-Phenyl-Trost system, there are totally five transition structures: **TS'_{left_Si_a/b}**, **TS'_{left_Re_a/b}** and **TS'_{right_Si}**. Thus, we have revised Scheme 5 and Scheme 6a as follows.

Scheme 5. DFT-optimized structures and relative energies of stereo-Determining C–C bond formation transition states of the reaction of **1b** with **2a** catalyzed by the **DACH-ZYC-Phos-C1**-ligated palladium catalyst. Hydrogen-bonding interactions are shown as red dash lines; selected bond lengths (Å) and bond angles (°) are listed.

Scheme 6. a. DFT-optimized structures and relative energies of ee determining C–C bond formation transition states for the reaction of **1b** with **2a** catalyzed by the DACH-Phenyl-Trost-ligated palladium catalyst. Hydrogen-bonding interactions are shown as red dash lines; selected bond lengths (Å) are listed.

For issue c: In this reaction, the methylene- π -allyl is not a prochiral reactant. Thus, we only described the approaching faces of **2a** anion as *Re* or *Si*.

3. In addition to the steric interactions and the importance of the hydrogen bond, the authors should show the distances of the bonds formed in transition states.

Response: Thanks for your comments. We have added the distances of the forming bonds in the

transition structures in the revised Scheme 5 and Scheme 6a, as you suggested.

4. Taking into account that the authors use N-Ph pyrazolones, it would be of interest to analyze the contribution of the p π i interactions.

Response: Thanks for your suggestion. In this study, N-phenyl substituted substrates have been applied. In order to clarify the suggested issue, we have conducted the reactions of the N-alkylpyrazolones substrates, *N*-cyclohexyl pyrazol-5-one **2u** and *N*-benzyl pyrazol-5-one **2v**, and added the results to entries 27 and 28 of Table 3

Table 3. The Scope of allenes and pyrazol-5-ones. ^a

entry	1	R ² /R ³ /R ⁴ (2)	yield (ee) of 3 (%)
1	1a	Me/Bn/Ph (2a)	96 (95, (S)- 3aa)
2	1b	Me/Bn/Ph (2a)	91 (93, (S)- 3ba)
3	1c	Me/Bn/Ph (2a)	90 (93, (S)- 3ca)
4	1d	Me/Bn/Ph (2a)	96 (94, (S)- 3da)
5	1e	Me/Bn/Ph (2a)	>99 (97, (S)- 3ea)
6 ^b	1f	Me/Bn/Ph (2a)	86 (78, (R)- 3fa)
7	1a	Me/ o -MeC ₆ H ₅ CH ₂ /Ph (2b)	97 (95, (R)- 3ab)
8	1a	Me/ m -MeC ₆ H ₅ CH ₂ /Ph (2c)	96 (92, (S)- 3ac)
9	1a	Me/ p -MeC ₆ H ₅ CH ₂ /Ph (2d)	92 (95, (S)- 3ad)
10	1a	Me/ p -ClC ₆ H ₅ CH ₂ /Ph (2e)	97 (95, (S)- 3ae)
11	1a	Me/ p -BrC ₆ H ₅ CH ₂ /Ph (2f)	96 (94, (S)- 3af)
12	1a	Me/ p -FC ₆ H ₅ CH ₂ /Ph (2g)	94 (94, (S)- 3ag)
13	1a	Me/ p -NCC ₆ H ₅ CH ₂ /Ph (2h)	97 (94, (S)- 3ah)
14	1a	Me/ p -CF ₃ C ₆ H ₅ CH ₂ /Ph (2i)	96 (94, (S)- 3ai)
15	1a	Me/ m -MeOC ₆ H ₅ CH ₂ /Ph (2j)	95 (94, (S)- 3aj)
16	1a	Me/ m -O ₂ NC ₆ H ₅ CH ₂ /Ph (2k)	86 (98, (S)- 3ak)
17 ^c	1a	Me/Et/Ph (2l)	87 (94, (S)- 3al)
18 ^d	1a	Me/allyl/Ph (2m)	96 (93, (S)- 3am)
19 ^e	1a	Me/allyl/Ph (2m)	95 (94, (R)- 3am)
20	1a	Me/ α -NaphthylCH ₂ /Ph (2n)	95 (97, (R)- 3an)
21	1a	Me/CH ₂ CH ₂ OH/Ph (2o)	94 (80, (R)- 3ao)
22	1a	Me/CH ₂ CH ₂ OTBS/Ph (2p)	91 (93, (R)- 3ap)
23 ^{f,g}	1a	Me/Ph/Ph (2t)	62 (36, (R)- 3at)
24	1a	Et/Bn/Ph (2q)	>99 (96, (S)- 3aq)
25	1a	ⁱ Pr/Bn/Ph (2r)	91 (97, (S)- 3ar)
26	1a	Ph/Bn/Ph (2s)	94 (95, (S)- 3as)

27 ^f	1b	Me/Bn/Cy (2u)	84 (95, (S)- 3bu)
28 ^f	1b	Me/Bn/Bn (2v)	82 (90, (S)- 3bv)

^a Reaction conditions: **1** (0.5 mmol), **2** (1.2 equiv), Pd₂(dba)₃•CHCl₃ (2.5 mol%), and **DACH-ZYC-Phos-C1** (5.0 mol%) in toluene (25 mL) at 60 °C unless otherwise noted. The ee of **3** was determined by chiral HPLC. ^b The reaction time was 2 h. ^c **2I** (1.4 equiv) was used and the reaction time was 13 h. ^d The reaction time was 11 h. ^e *ent*-**DACH-ZYC-Phos-C1** was used and the reaction time was 11 h. ^f Pd₂(dba)₃•CHCl₃ (3.0 mol%), and **DACH-ZYC-Phos-C1** (6.0 mol%) at 80 °C. ^g Recovery of **1a** was 9% as determined by the ¹H NMR analysis of the crude product using mesitylene as the internal standard.

Thus, we could exclude the influence of the π/π interactions between the aromatic rings of the ligand and N-Ph pyrazolones on the enantioselectivity. Furthermore, we also have tried to analyze the non-covalent interactions (NCI) in TSs using the Multiwfn (*J. Comp. Chem.* **2012**, 33, 580.) and VMD programs. No obvious π/π or dispersive attractions are found so far from the analysis. The plot of non-covalent interactions of **TS_left_Si_a** are shown as follows for example. And we have added this figure to Supplementary Information (Supplementary Figure 3).

The NCI plot of **TS_left_Si_a**

5. It is recommended not to use the descriptors *endo* and *exo*

Response: Thanks for your comments. In this manuscript, *endo* and *exo* were labeled according to Helmchen's conventional notation specifying the orientation of its allyl fragment with respect to the metal-center plane. These descriptors (*endo/exo*) are widely used in reactions with π -allyl palladium complexes participation. For selected literatures as follows: *J. Am. Chem. Soc.* **2009**, 131, 9945–9957; *Chem. Eur. J.* **2011**, 17, 2885 – 2896; *Eur. J. Org. Chem.* **2016**, 4800–4804; *ACS Catal.* **2018**, 8, 3587–3601.

6. the manuscript must be thoroughly revised to avoid typographical errors

Response: Thanks for your suggestion. We have checked the manuscript carefully.

Few precedents exist in the literature for the stereoselective functionalization of pyrazolones

Response: Thanks for your positive comments.

REVIEWERS' COMMENTS

Reviewer #1 (Remarks to the Author):

The revised manuscript should be accepted after careful the authors' re-correction.

Reviewer #2 (Remarks to the Author):

The authors have properly revised their manuscript, which has become acceptable for publication. I have no further comments.

Reviewer #3 (Remarks to the Author):

The authors have taken into accounts all suggestions to improve the manuscript quality. Thank you very much for the efforts
Accepted to publish